# Sonographic Detection of Fetal Cholelithiasis

**DOI:** 10.3390/diagnostics13182900

**Published:** 2023-09-11

**Authors:** Nicolae Gică, Andra Radoi, Corina Gică, Anca Maria Panaitescu, Gheorghe Peltecu, Iulia Huluță

**Affiliations:** 1Clinical Hospital of Obstetrics and Gynaecology Filantropia, 011132 Bucharest, Romania; mat.corina@gmail.com (C.G.); anca.panaitescu@umfcd.ro (A.M.P.); gheorghe.peltecu@umfcd.ro (G.P.); iulia.huluta@drd.umfcd.ro (I.H.); 2Obstetrics and Gynecology Department, Faculty of Medicine, Carol Davila University of Medicine and Pharmacy, 050474 Bucharest, Romania; andra-ioana.radoi@rez.umfcd.ro

**Keywords:** fetal cholelithiasis, cystic fibrosis, third-trimester ultrasound, gallbladder

## Abstract

Fetal biliary lithiasis is a benign condition characterized by the presence of gallstones in the gallbladder of a developing fetus. It is typically detected incidentally during a routine obstetric echography. The incidence of this condition varies from 0.03% to 2.3%. In most cases, fetal cholelithiasis resolves spontaneously and has an excellent prognosis. However, there are certain risk factors that may contribute to its development. Maternal factors that increase the risk of fetal cholelithiasis include placental abruption, elevated estrogen levels, narcotic use, diabetes, enteral nutrition, and specific medications, such as ceftriaxone, furosemide, and prostaglandin E2. Fetal factors that can contribute to the condition include Rhesus or ABO blood group incompatibility, congenital anomalies affecting the cardiovascular, gastrointestinal, or urinary systems, twin pregnancies with the fetal demise of one twin, genetic anomalies such as trisomy 21, chromosomal aberrations, cystic fibrosis, growth restriction, oligohydramnios, hepatitis, or idiopathic causes. Usually, the gallstones spontaneously resolve before or after birth without requiring specific treatment. However, in rare instances, complications can arise, such as the formation of biliary sludge, inflammation of the gallbladder (cholecystitis), or obstruction of the bile ducts. If complications occur or if the gallstones persist after birth, further evaluation and management may be necessary. Treatment options can include medication, minimally invasive procedures, or, in severe cases, surgical removal of the gallbladder.

**Figure 1 diagnostics-13-02900-f001:**
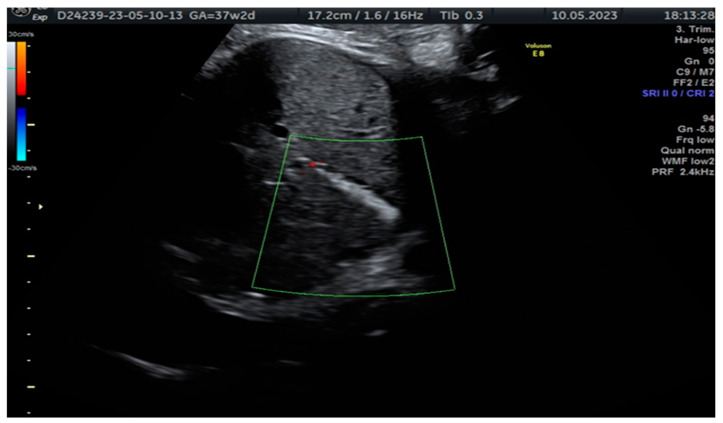
In the case reported, a 34-year-old woman, pregnant for the first time and with no significant medical history or consanguineous marriage, was referred to our fetal medicine department at 37 weeks of gestation for an intraabdominal hyperechoic image assessment. She was Resus-positive, and CMV-immune. Upon ultrasound examination, we identified a hyperechoic tubular structure at the level of the gallbladder without any additional malformation. The patient was informed about the diagnosis of fetal cholelithiasis and its potential implications. She had already been counselled about the association with cystic fibrosis and had the CFTR gene mutation tested. She was not a carrier of the mutation. The image is a cross-sectional view of the fetal abdomen in 2D ultrasound, below the level of the stomach, showing a hyperechoic tubular structure in the fetal abdomen with no colour Doppler signal. This is a highly suggestive image of fetal cholelithiasis. Meconium peritonitis was considered within the spectrum of potential diagnoses; this condition occurs when the fetus experiences intrauterine bowel perforation, leading to the accumulation of meconium (the first stool) within the abdominal cavity. It can cause calcifications within the abdomen. However, it was ultimately excluded as a plausible aetiology due to the absence of concomitant findings, such as elevated amniotic fluid volume (polyhydramnios) or bowel dilation. Fetal cholelithiasis can be caused by fetal hepatitis (CMV infection), haemolytic disorders, polycythemia and biliary atresia, or the condition can be idiopathic [1,2,3,4]. Furthermore, biliary disorders, including gallbladder and bile duct disease, are common complications of cystic fibrosis. Therefore, genetic consultation and proper testing for the parents and the fetus are highly recommended after the detection of abdominal hyperechoic structures [1]. The diagnosis was confirmed by abdominal ultrasound but resolved by the 2-month follow-up. Fortunately, there were no additional malformations detected during the scan, and the baby did not experience jaundice or any other complications during the neonatal period. This positive outcome suggests that the gallstones were a transient issue and that the baby is on a healthy developmental trajectory.

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
