# Peer review of "Sonographic Detection of Fetal Cholelithiasis"

_diagnostics, 2023, doi:10.3390/diagnostics13182900_

Round 1
Reviewer 1 Report
This is an interesting image of fetal colelithiasis. It is also an interesting reminder of the importance of cystic fibrosis screening after detection of abdominal hyperechoic structures.
Author Response
Dear Reviewer,
Thank you for your comments.
Best regards,
Reviewer 2 Report
Interesting and rare article. The picture is appropriate. Title, presentation is decent.
Author Response

(The authors gave the same response as above.)

Reviewer 3 Report
Congratulations on your work on this case of sonographic presentation of fetal cholelithiasis. Here I would like to offer my suggestions aimed at improving your manuscript.
In the context of differential diagnosis, you refer to the meconium peritonitis, as a possible clinical aspect of this case. However, you provide no further explanation of the sonographic similarities among these two clinical entities, or the reason you gravitate towards fetal cholelithiasis as the most likely diagnosis among these two.
The last paragraph describing the figure, should be more structured. Patient's medical history, followed by imaging findings, laboratory testing, differential diagnosis and further recommendations should me clearly distinguished.
Overall correct use of the English Language.
Minor corrections needed.
Author Response
Dear Reviewer,
Thank you for your comments.
The last sentence was reviewed "Therefore, genetic consultation and proper testing for the parents and the fetus are highly recommended after the detection of abdominal hyperechoic structures.[1] The diagnosis was confirmed by abdominal ultrasound but resolved by the 2-month follow-up. Fortunately, there were no additional malformations detected during the scan, and the baby did not experience jaundice or any other complications during the neonatal period. This positive outcome suggests that the gallstones were a transient issue, and the baby is on a healthy developmental trajectory"
Reviewer 4 Report
The authors reported the case of a 34-year-old woman who was pregnant for the first time and had no special medical history or consanguinity. The patient was referred to the fetal medicine department at 37 weeks' gestation for an intra-abdominal hyperechoic image at the level of the gallbladder, which was eventually diagnosed as fetal cholelithiasis.
This case is poorly presented and not informative.
An abstract is poorly structured and repeats known facts from the literature without even a word about the present case. The present case should be described in detail in the abstract, whereas the facts about fetal cholelithiasis should be reduced to a minimum.
Further presentation of the case is poor. More details about the patient should be added.
The ultrasound report is also poor, a more detailed description of the sonographic findings should be added (volume/dimensions of the mas, possible dimensions and number of stones...).
The patient's post-delivery care was not even mentioned. How long the patient was followed up after delivery, what happened to the stones after delivery, also whether cholelithiasis was confirmed after delivery.
Finally, this report is poorly written and inadequate. In addition, this condition is well known and has been described several times in the literature. There is no new information in this report (interesting image ).
The quality of the English language should be improved. The manuscript should be edited by a native English speaker or a professional language editor to improve grammar and readability.
The quality of the English language should be improved. The manuscript should be edited by a native English speaker or a professional language editor to improve grammar and readability.
Author Response
Dear Reviewer,
Thank you for your comment and your attention to these important post-delivery care and follow-up aspects. We took your feedback into consideration and have made the necessary additions to the text to address these points.
We acknowledge the importance of clear and structured communication in presenting medical cases. To address this, we have reorganized the last paragraph.
We appreciate your insights regarding the writing quality and the familiarity of the condition discussed.
Our article is not a case report, it is a short article - interesting images type.
The article was reviewed by a narrative English speaker.
Best regards,
Round 2
Reviewer 4 Report
The authors have made some corrections, but I cannot see any significant improvement. Although this report is an 'interesting image' it should be reported with more details regarding the clinical presentation and US findings. In addition, as I mentioned earlier, fetal cholelithiasis is well known and has been described several times in the literature. There is no new information in this report (interesting image) and I do not believe that this report deserves publication in a high impact factor international journal.
Moderate editing of English language required